# Mutations Affecting Genes in the Proximal T-Cell Receptor Signaling Pathway in Peripheral T-Cell Lymphoma

**DOI:** 10.3390/cancers14153716

**Published:** 2022-07-29

**Authors:** Xiaoqian Liu, Jinyao Ning, Xuxiang Liu, Wing C. (John) Chan

**Affiliations:** 1Department of Hematology, Affiliated Yantai Yuhuangding Hospital, Qingdao University, Yantai 264000, China; liuxiaoqian83@126.com; 2Department of Thyroid Surgery, Affiliated Yantai Yuhuangding Hospital, Qingdao University, Yantai 264000, China; docning@126.com; 3Department of Pathology, City of Hope National Medical Center, Duarte, CA 91010, USA; xuxli@coh.org

**Keywords:** peripheral T-cell lymphoma (PTCL), TCR signaling pathway, mutation, fusion protein

## Abstract

**Simple Summary:**

The advent of next-generation sequencing (NGS) has allowed rapid advances in genomic studies on the pathogenesis and biology of peripheral T-cell lymphoma (PTCL). Recurrent mutations and fusions in genes related to the proximal TCR signaling pathway have been identified and show an important pathogenic role in PTCL. In this review, we summarize the genomic alterations in TCR signaling identified in different subgroups of PTCL patients and the functional impact of these alterations on TCR signaling and downstream pathways. We also discuss novel agents that could target TCR-related mutations and may hold promise for improving the treatment of PTCL.

**Abstract:**

Peripheral T-cell lymphoma (PTCL) comprises a heterogeneous group of mature T-cell malignancies. Recurrent activating mutations and fusions in genes related to the proximal TCR signaling pathway have been identified in preclinical and clinical studies. This review summarizes the genetic alterations affecting proximal TCR signaling identified from different subgroups of PTCL and the functional impact on TCR signaling and downstream pathways. These genetic abnormalities include mostly missense mutations, occasional indels, and gene fusions involving CD28, CARD11, the GTPase RHOA, the guanine nucleotide exchange factor VAV1, and kinases including FYN, ITK, PLCG1, PKCB, and PI3K subunits. Most of these aberrations are activating mutations that can potentially be targeted by inhibitors, some of which are being tested in clinical trials that are briefly outlined in this review. Finally, we focus on the molecular pathology of recently identified subgroups of PTCL-NOS and highlight the unique genetic profiles associated with PTCL-GATA3.

## 1. Introduction

Peripheral T-cell lymphoma (PTCL), as defined by the World Health Organization (WHO) classification, comprises a heterogeneous group of mature T-cell malignancies, most of which are associated with poor clinical outcomes [1]. Knowledge of the pathogenesis and biology of PTCLs lags far behind its B-cell counterparts. More recently, the advent of next-generation sequencing (NGS) has allowed rapid advances in genomic studies that have greatly improved our understanding of PTCL. Similar to other lymphomas, mutations of genes that regulate the epigenome are frequent in PTCL, and loss of function mutations in the *TET2* gene is likely to be a founder mutation in angioimmunoblastic lymphoma (AITL). T-cell receptor (TCR) signaling is critical in T-cell activation and survival, and perturbation of this pathway is an important pathogenic mechanism of PTCL. Recurrent activating mutations in genes related to the proximal TCR signaling pathway have been identified, and new therapies targeting the TCR pathway are being investigated in preclinical and clinical studies [2,3]. Abnormalities affecting more downstream signaling pathways are prominent in certain PTCLs, such as JAK/STAT3 signaling in anaplastic large cell lymphoma (ALCL) and *STAT5B* mutations in monomorphic epitheliotropic intestinal T-cell lymphoma (MEITL) and hepatosplenic T-cell lymphoma (HSTCL). In this review, we will focus on genetic alterations detected in PTCL affecting the proximal TCR signaling pathway and discuss the pathogenetic mechanisms and novel drugs involved.

## 2. TCR Signaling Pathways

The TCR is a complex of integral membrane proteins which in most T cells is composed of an α and a β chain (only about 5% of T cells carry the γδ-TCR), recognizing MHC-presented peptide antigens (pMHC). When the TCR αβ subunit interacts with an MHC-peptide complex with sufficient affinity, the signal is transmitted through the associated CD3 complex consisting of hetero-dimers of γ, δ, and ε, and homo-dimers of ζ, carrying a total of 10 immunoreceptor tyrosine-activation motifs (ITAM) that can be tyrosine phosphorylated with conformational alterations upon the engagement of the TCR αβ/pMHC. CD4 or CD8 coreceptors may associate with MHC class II and I molecules, respectively, and promote CD3 complex activation. The activated Src-family kinase, cytoplasmic lymphocyte-specific protein tyrosine kinase (LCK), phosphorylates the tyrosines in the ITAM of the CD3 ζ chains (and to a lesser extent, those on the other CD3 subunits) that serve as binding sites for the downstream ζ chain associated protein kinase 70 (ZAP70), which is then phosphorylated and activated by LCK. Activated ZAP70 phosphorylates adapter proteins such as the linker for activation of T cells (LAT) and SH2 domain-containing leukocyte phosphoprotein of 76 kDa (SLP76), thus facilitating the formation of the LAT signalosome [4] composed of LAT, SLP-76, GADS, GRB2, phospholipase C γ1(PLCγ1), interleukin-2-inducible T-cell kinase (ITK), Guanine nucleotide exchange factor 1 (VAV1), SOS1 (an activator of the RAS-MEK-ERK pathway), ADAP, and SKAP1. ITK phosphorylates and activates PLCγ1, which then cleaves phosphatidylinositol (4,5) bisphosphate to generate two important second messengers, inositol trisphosphate (IP3) and diacylglycerol (DAG).

IP3 binds to receptors on the endoplasmic reticulum (ER), leading to an initial phase of calcium release that is critical in the activation of nuclear factor of activated T cells (NFAT). The proximal signaling events thus lead to the downstream activation of critical pathways including the mitogen-activated protein (MAP) kinases ERK and p38, the transcription factors nuclear factor kappa B (NF-κB), and NFAT [5,6].

The binding of CD28 to CD80/CD86 expressed by antigen-presenting cells (APCs) provides a second signal that is required for full T-cell activation. Phosphorylated Tyr residues of the Tyr-Met-Asn-Met (YMNM) motif on CD28 bind directly to the p85 regulatory subunit of phosphatidylinositol 3-kinase (PI3K), whose catalytic subunit can catalyze the generation of phosphatidylinositol 3,4,5-trisphosphate (PIP3) from phosphatidylinositol 4,5-biphosphate (PIP2). PIP3 contributes to the membrane recruitment of several important proteins, including phosphorylated 3-phosphoinositide-dependent protein kinase-1(PDK1), which activates a number of important signal transducers, including protein kinase B (PKB/Akt), p70 ribosomal protein S6 kinase (p70S6K), cyclic AMP-dependent protein kinase and PKCθ. PKCθ subsequently phosphorylates membrane-associated CARD11, which then forms a complex with BCL10 and MALT1 (the CBM complex) [7]. The CBM complex, through several intermediate steps, activates nuclear factor κB (NF-κB) and the JNK pathway [8]. The Asn residue of the YMNM motif is thought to be involved in the phosphorylation of VAV1 via GRB2/GADS and the GRB2-VAV1-JNK pathway. The activation of ITK depends mainly on the PRRP and PYAP regions [9,10,11] (Figure 1 and Figure 2).

## 3. Chronic Antigenic Stimulation and Persistent TCR Signaling in PTCLs

Clinical observation suggests that some PTCLs are preceded by polyclonal or oligoclonal T-cell populations associated with a permissive cytokine milieu and TCR stimulation by exogenous or autoantigens. Long-lasting clonal evolution may precede neoplastic transformation. Examples are the presence of chronic dermatitis before mycosis fungoides(MF)/Sézary syndrome(SS) [12], gluten-sensitive enteropathy preceding enteropathy-associated T-cell lymphoma (EATL) [13], and autoimmune cytotoxic T-cell expansions prior to the emergence of T-cell large granular lymphocyte leukemia (T-LGL) [14,15]. Hepatosplenic T cell lymphoma (HSTL) [16] and breast implant-associated ALCL [17] also seem to occur in a setting of sustained immune stimulation.

The mechanisms of T cell lymphoma development in the context of chronic TCR stimulation were investigated using conventional *p53*^−/−^ T cells [18]. Chronic TCR stimulation triggered PTCL development and promoted epigenetic T cell reprogramming toward NK-like cells, downregulating several T cell-specific genes, such as *Bcl11b* and inducing several NK cells associated features, such as NK cell receptors (NKRs) and their signaling molecules. This NK-like reprogramming induced addiction to SYK and NK cell-activating receptors (NKaR) signaling to maintain PTCL survival, whereas TCR signaling was mostly ineffective. Thus, in certain situations, as illustrated in this study [18], chronic TCR signaling, despite being important for lymphomagenesis, may not be evident in PTCL. This is observed clinically by the loss of TCR expression in some PTCL, as reported in AITL [19] and particularly in ALCL [20]. It is possible that signaling could be substituted by abnormal ALK activity due to translocation or mutations in the TCR signaling pathway may alleviate the requirement for signaling through the TCR [20,21,22].

## 4. Activating Mutations in Genes Related to TCR Signaling in PTCLs

In recent years, high throughput sequencing on PTCLs has revealed mutations in genes related to proximal TCR signaling. These alterations involve mostly missense mutations but also small indels, genomic copy number alterations (gCNAs), and fusions involving *RHOA*, *VAV1*, *FYN*, *LCK*, ITK, *PLCG1*, *PRKCB*, *CARD11*, *CD28*, and PI3K subunits, whose frequencies vary with different subtypes of PTCLs [2,17,23,24,25,26,27].

### 4.1. RHOA and VAV1

The Ras homolog family member A (*RHOA*) gene is one of the most frequently mutated genes with its mutations detected in 50–71% of AITL [2,25,26], 8–18% of PTCL-NOS [25], 5% of cutaneous T-cell lymphomas (CTCL) [28], and 8% of adult T cell leukemia/lymphoma (ATLL) [23]. RhoA belongs to the Rho family of GTPases and involves in cytoskeleton remodeling, cell morphology, migration, signaling, proliferation, and survival [29]. RhoA protein activation is controlled by Guanine Nucleotide Exchange Factors (GEF) and GTPase activating proteins (GAP), which promote the switch of the GTPase from a GTP-activated state to a GDP-inactivated state and vice versa. VAV1 is a multi-functional domain protein with GEF activity encoded by the *VAV1* gene, where mutations and fusions were also observed quite frequently in PTCL [2].

A highly recurrent mutation affecting the GTP/GDP binding domain, *RHOA^G17V^*, was identified as the predominant variant and is particularly related to follicular helper T (Tfh) related PTCL. Although the mutant RHOA G17V lacks GTP binding capacity, suggesting defects in classical RHOA signaling, and is believed to be a dominant negative (DN) mutant, how this mutant contributes to PTCL pathogenesis is not entirely resolved. In B-cell lymphomas, *RHOA* mutations are rarely observed, but mutations affecting the upstream molecules, such as GNA13, that negatively affect RHOA functions, are moderately frequent. These results in altered migration of the B-cells and activation of the PI3K pathway due to decreased activity of the ROCK kinases with lowered PTEN function [30]. Simultaneously, there is increased RAC pathway activation. It would be interesting to explore whether similar functional alterations occur in PTCL with the *RHOA^G17V^* mutation. It has been reported that RHOA G17V binds to VAV1, augmenting its adaptor function through phosphorylation of 174Tyr, resulting in acceleration of TCR signaling, enrichment of cytokine and chemokine-related pathways, and Tfh polarization [19,31]. However, VAV1 is not the preferred GEF associated with RHOA and over-expression of RHOA G17V in the in vitro experimental system may alter protein associations. It has been reported that mutant compared to wild-type RHOA exhibited enrichment of the RAC1 pathway as well as T-cell cytokine signaling, NOTCH, and NF-kB pathways [32].

*RHOA^G17V^* expression together with *Tet2* loss resulted in the development of an AITL-like tumor in mice [19,33]. In vitro experiments revealed a dominant negative role for the *RHOA^G17V^* mutation that induced Tfh specification, increased proliferation associated with ICOS upregulation, and increased PI3K and MAPK signaling [19].

Other *RHOA* mutations (Table 1), albeit far less frequent than G17V, have been reported in non-Tfh PTCL, affecting amino acids 16–19 (C16R, G17E, K18N, and T19I) or amino acids 117–118 (N117I/K, K118E/T) of the GTP/GDP binding domains, or affecting other regions with non-validated function. Some of these mutations appear to be activating mutations with gain of function [2,28]. The K18N mutation occurs in the highly conserved GTP binding site of RHOA and shows a marked increase in binding affinity to the GTP. Moreover, unlike RHOA G17V, RHOA K18N showed a marked increase in luciferase activity from a reporter with serum-response elements under serum activation [2]. The specific role of these mutations in PTCL pathogenesis needs further study.

*RHOA^G17V^* could be detected in biopsies prior to the lymphoma diagnosis (0–26.5 months, mean 7.87 months) and could be valuable in the early detection of AITL and PTCL-Tfh [34]. Clinically, *RHOA^G17V^* -mutated patients had a significantly higher incidence of splenomegaly and B symptoms at diagnosis, and exhibited more classical morphology, a higher positive rate for CD10, BCL6, and CXCL13 expression, and a stronger PD1 expression, but there was no significant difference in overall survival (OS) between mutated and wild-type subgroups in most of the studies [35,36,37]. While one study reported poor OS and progression-free survival in *RHOA^G17V^* -mutated AITL [38], another study revealed that *RHOA* mutation-negative cases had shorter relapse-free survival and showed a trend towards shorter overall survival [39].

The aberrant signaling induced by RHOA G17V was reported to be efficiently inhibited by dasatinib or a pan-Src inhibitor PP2 in ex vivo experiments. [31] Dasatinib was applied in several clinical trials enrolling relapsed/refractory T cell lymphoma (NCT00608361, NCT00550615, and NCT01609816). In one of the trials (NCT00550615), the objective response rate was 29.2%. It is important to conduct clinical studies to evaluate the efficacy of dasatinib with well-designed correlative studies to determine if there is any association with AITL patients with aberrant RHOA-VAV1 activation or other specific mutations.

VAV1 is a hematopoietic-specific GEF and a primary activator of RAC in T cells. VAV1 integrates signals from tyrosine kinases to activate RAC, and potently regulates lymphocyte development and activation. Genetic knockout of *VAV1* results in dramatic defects in TCR signaling and clustering, reduced calcium flux, and impaired NFAT transcriptional activity [40,41].

*VAV1* translocations and missense/deletion mutations (shown in Figure 3) have been identified in T-cell lymphoma [2,31,42,43]. Frequencies of *VAV1* alterations are 18% in ATLL [23], 11% in ALCL, 7% to 11% in PTCL-NOS, and 5% in AITL [31,42,44,45]. Missense mutations and focal in-frame deletions accumulate in the Ac (Glu175, D165–174) and SH3 (Arg798, Arg822, and D778–786) domains. VAV1 fusion, while involving many partner genes, including *VAV1-THAP4*, *VAV1-MYO1F*, *VAV1-S100A7*, *VAV1-STAP2*, and *VAV1-GSS* [43], all results in deleting the VAV1 C-terminal SH3 domain. Alterations in either the Ac or SH3-SH2-SH3 domains impair their interaction with the Dbl homology (DH) domain, disrupting VAV1 autoinhibition [46]. Both alterations in Ac and SH3 domains lead to hyperactivation of TCR signaling in vitro [31,44], though how substitutions in the PH or ZF domains perturb downstream signaling remains unclear. For functional analysis, spontaneous activation of TCR signaling and subsequently increased NFAT and NF-κB activity were observed in Jurkat cells expressing VAV1 fusion protein [47]. Mouse models with *VAV1(del165-174)* and *VAV1-STAP2* have been developed, but they did not lead to malignant transformation of T cells in vivo. Mature T cell neoplasms (TCNs) were only observed in *p53^−/−^ VAV1* deletion/fusion mice. Both VAV1 deletion (D165-174) and fusion (VAV1-STAP2) mutants have similar oncogenic properties because tumors harboring either of these two mutants display similar phenotypes, gene expression profiles, and genomic abnormalities [43].

Mogamulizumab, a defucosylated anti-CCR4 (chemokine receptor 4) antibody, was reportedly effective against some PTCLs that express CCR4 on tumor cells [48]. Given the high CCR4 expression observed in VAV1-mutant tumors and the positive correlation between CCR4 and GATA3 expression in human PTCL-GATA3 [49], PTCL with VAV1 mutations should be evaluated for CCR4 expression as mogamulizumab could be a treatment option [43]. JQ1, which is a bromodomain inhibitor that targets the Myc pathway, was reported to prolong the OS of mice harboring VAV1-mutant tumors [43].

### 4.2. FYN, LCK, and ITK

The tyrosine kinases encoded by *FYN* and *LCK* are the predominant SRC family kinases found in T lymphocytes. While LCK plays a critical role in T-cell activation upon T-cell receptor (TCR) stimulation [50], the role of FYN is less well defined, but mutations affecting *FYN* are far more common. *FYN* mutations, including L174R, R176C, and Y531H, are detected in PTCL-NOS and AITL with a frequency of 4% in the entire PTCL cohort [25]. FYN L174R and R176C specifically disrupt the intramolecular inhibitory interaction of the FYN SH2 domain with FYN Tyr531 residue [25]. In ex vivo experiments, *FYN^L174R/R176C/Y531H^* were confirmed as activating mutations with abrogation of the auto-inhibitory loop and increased levels of FYN phosphorylation compared with control cells expressing wild-type *FYN*. In a cohort of AITL and other Tfh-cell-derived lymphomas, five mutations occurred in three patients. One patient harbored 2 mutations, S186L affecting the SH2 domain and T524fs leading to the absence of phosphorylated tyrosine 531, which probably conferred enhanced kinase activity by disrupting their inhibitory interaction [2]. Two mutations (E107S and K108fs) in adjacent positions in the SH3 domain were observed on the same allele in another patient. A Q527X mutation affecting Tyr531 phosphorylation was detected in the third patient. *FYN* mutations (F521fs and Q527X) affecting the SH3 domain were identified in ATLL with an overall frequency of 4% [23].

*FYN-TRAF3IP2* as a recurrent oncogenic gene fusion was identified with a frequency of 23% in PTCL, and most of the fusions occurred in AITL and PTCL-NOS [51]. This high frequency of fusion needs to be validated by additional studies with a large number of patients. *FYN-TRAF3IP2* leads to aberrant NF-κB signaling downstream of TCR activation, indicating its role as a driver oncogene [51]. In vitro treatment of FYN-TRAF3IP2 mouse tumor cells with the IKK inhibitors BMS-345541 [52] and IKK-16 [53] induced strong dose-dependent anti-lymphoma effects [51].

LCK mutations in the tyrosine kinase domain (N446K, P447R) were reported in a single AITL patient. Although predicted to be activating, their functional characteristics have not yet been determined. [2]

ITK is one of the Tec family kinases that plays a critical role in all stages of T cell development. The recurrently detected translocation t(5;9) (q33;q22) leads to the fusion of *ITK* with *SYK*. *ITK-SYK* transcripts were initially reported in 17% of PTCL-NOS, but not in the cases of AITL and ALK^−^ ALCL [54]. Most of the cases with *ITK-SYK* were later confirmed as Tfh like-PTCL [55]. ITK-SYK proteins were demonstrated in Jurkat cells to activate the IL2RG/JAK3/STAT5 signaling pathway, which was associated with the abundant secretion of IL-2 and IL-21 [56]. In a mouse model of *ITK-SYK* induced T-cell lymphoma, *ITK-SYK* was found to enhance the expression of PD-1 that subsequently attenuated the AKT and PKC activities in premalignant *ITK-SYK* expressing cells [57]. Elimination of this immune checkpoint markedly accelerated tumor development, indicating that PD-1 can serve as a tumor suppressor gene in certain situations. It is interesting to note that PD-1/PD-L1 blockade in patients with PTCL has led to hyperprogression of the tumor [57]. An in-frame *ITK-FER*(Feline Encephalitis Virus-Related kinase) fusion was detected in PTCL-NOS, leading to increased colony formation in HEK-293T cells [42] and enhancement of tyrosine kinase activity and STAT3 phosphorylation [58].

CPI-818, an oral ITK inhibitor, is studied in a Phase 1/1b, open-label, first in human clinical trial for the treatment of relapsed/refractory T-cell lymphoma (NCT03952078). This trial will examine the safety, tolerability, and anti-tumor activity of CPI-818 as a single drug. The dual SYK/JAK inhibitor, Cerdulatinib, demonstrated good clinical response in relapsed/refractory PTCL patients in a phase 2 clinical trial [59]. While the profile of response did not appear to depend on SYK rearrangement, perhaps cases with aberrant SYK activation would be particularly sensitive.

### 4.3. PLCG1, PRKCB, and CARD11

*PLCG1*, which encodes Phospholipase C, gamma 1(PLCγ1), a key regulator of proximal TCR signaling, is one of the most frequently mutated genes in PTCL [2,23,60,61]. A *PLCG1* mutation is detected in 36% of patients in a cohort of 426 ATLL patients [23]. *PLCG1* exhibits several hotspot missense mutations, such as R48W, S345F, S520F, E1163K, and D1165H, enhancing the activities of PLCγ1 and activation of downstream NFAT and NF-κB pathways [23,24,60]. In AITL and other Tfh-cell-like lymphomas, missense mutations in the coding region of *PLCG1* were detected in 14.1% of 85 patients [2], in which two novel variants (E730K and G869E) were identified. The activity of all these variants in *PLCG1* was investigated in vitro, and the experiments confirmed that the S345F, S520F, and other variants including D342G in PI-PLC, E730K in SH2, G869E in SH3, and E1163K and D1165G/H in C2 domains were activating mutations and promoted mucosa-associated lymphoid tissue lymphoma translocation protein 1(MALT1) cleavage and NFAT activity [2]. Of the two variants in the PH1 domain, E47K, but not R48W, was confirmed as an activating mutation [2]. Reports on *PLCG1* mutations in CTCL are contradictory. In the study by Vaqué et al. [60], 21% of 53 patients with CTCL were found to have *PLCG1* mutations, and 10 (19%) patients in the entire cohort had the S345F mutation in exon 11, with the other one being the S520F mutation. In a French cohort of CTCL patients [62], *PLCG1^S345F^* was only found in one patient with Sézary syndrome (SS, n = 39) and none of the patients with mycosis fungoides (MF, n = 37). In the most recent study, seven *PLCG1* mutations in tumor cells were identified from 11 SS patients, including four recurrent aberrations (R48W, D342N, S345F, and E1163K) and were confirmed as activating mutations that stimulated downstream NF-κB, NFAT, and AP-1 transcriptional activity [63]. The frequency of the *PLCG1* mutation was analyzed in several published studies, with mutation frequencies of 15.1% in MF and 11.3% in SS [28]. Copy number variation of *PLCG1* is also detected with a frequency of 5% [28].

While *PLCG1* mutation is frequently detected in ATLL, AITL, CTCL, and PTCL-NOS, it is rarely identified in ALCL, EATL, and NKTCL [25,64,65,66].

*PRKCB* encodes a member of the protein kinase C (PKC) family of proteins (PKCβ), which acts downstream of PLCγ1 in TCR signaling [6]. In ATLL, PRKCB mutations occurred recurrently in the highly conserved regions within the catalytic domain with a prominent hotspot at Asp427(D427N/G/A) and is the second most frequent mutation with a frequency of 33% [23]. PKCβ is maintained in an inactive state through the autoinhibitory interaction between the C1b domain and the NFD helix [67]. It was expected that destabilizing the C1b clamp would promote its disengagement from the NFD helix and the rest of the catalytic domain, and therefore result in PKCβ activation [67]. Asp427 lies in close proximity to the essential residues (Tyr422 and Tyr 430) that stabilize the C1b-NFD interaction, while another hotspot mutation, Asp630, is located within the NFD motif. The stability of the C1b-NFD interaction is interrupted when the residues are altered by the hotspot mutations, resulting in PKCβ activation and the enhancement of the NF-κB pathway [23]. PRKCB mutations could also be detected in CTCL with a much lower frequency (0.4%) [28] compared to ATLL and are rarely reported in PTCL-NOS, AITL, ALCL, EATL, and NKTCL.

CARD11 is a cytoplasmic scaffold protein and member of the CBM complex, which is required for antigen receptor-induced NF-κB activation in both T- and B-cells [68]. In T cells, CARD11 is activated by TCR activated PLCγ1 and PKC family as well as CD28 mediated PI3K-AKT pathway. CBM complex mediates the activation of IKK and JNK and subsequent NF-κB and AP-1 activities, thereby promoting T-cell activation and survival [69]. Point mutations were reported in 24% of ATLL [23], 3.5% of AITL and Tfh like PTCL [2], and 5.4% of CTCL [28]. In T-cell lymphoma, *CARD11* mutations occur not only in the coiled-coil domain, as previously reported in B-cell lymphoma [70], but also affect the linker regions of the PKC-responsive auto-inhibitory domain, with a prominent hotspot mutation E626K [23]. E626K in ATLL leads to the deletion of the PKC-responsive auto-inhibitory domain and, consequently, constitutive activation of CARD11 and an increase in NF-κB activity [23,71].

The F176C mutation maps to the coiled-coil domain while the S547T and F902C variants detected in AITL and Tfh-like PTCL lie in the linker region. All three variants were confirmed to have gain-of-function properties that promoted the activation of NF-kB [2]. Besides point mutations, small intragenic deletions, and gene amplifications of *CARD11* are frequently detected in ATLL [23] and CTCL [28].

Notably, in ATLL, *CARD11* mutations were reported to exhibit a significant positive correlation with *PRKCB* mutations, suggestive of potential functional synergism between these concurrent lesions that were confirmed in ex vivo experiments [23].

MS-553 is a selective PKC inhibitor and is now being investigated in chronic lymphocytic leukemia in a phase I clinical trial (NCT03492125). Targeted therapy for *PRKCB* may be considerable for ATLL in the future [72]. The efficacy of a MALT1 inhibitor, JNJ-67856633, is under evaluation (NCT03900598) for B-cell lymphoma and it may also be effective for T-cell lymphoma with *PRKCB* or *CARD11* alterations.

### 4.4. CD28 and PI3K Elements

*CD28* mutations, fusions, and copy number variants (CNVs) are frequently detected in AITL [2,73], PTCL-NOS [73], ATLL [23], and CTCL [74,75].

In a study including AITL, PTCL-NOS, and ALK^−^ ALCL, *CD28* mutations are relatively frequent in AITL with an incidence of 11.4%, including several hotspots, such as T195P, D124V, and D124E [73]. *CD28-ICOS* fusions (5%) [73] were also detected. AITL cases with *CD28* mutations have inferior survival compared with *CD28*-WT cases. CD28 D124V was demonstrated to have a higher affinity for its ligand CD86 than CD28 WT [73], while CD28 T195P had a higher affinity for GRB2 and GADS/GRAP2 than CD28 WT [76]. These mutants may thus have gain of function and were shown to have alterations in transcription and higher NF-κB pathway activation [73,76].

*CTLA4-CD28* fusion was reported in a study as a highly frequent alteration in T-cell lymphomas (AITL, PTCL-NOS, NKTCL) with an overall frequency of 38% [77]. Such a high frequency has not been observed in other studies, and there are concerns regarding the reliability of the findings [78]. In ATLL, more *CD28* fusions were detected with a frequency of 7–8.8% than *CD28* mutations (2%, F511V, D124E/V, T195P/L) [23]. CTLA4-CD28 fusion converts inhibitory signals from CTLA4 engagement into activating signals by replacing the intracellular portion of CTLA4 with that of CD28 [77]. (Figure 4) In addition, considering that CTLA4 binds B7 ligands with a much higher affinity than CD28 [79], the fusion protein has an enhanced ability to promote T-cell activation. Blocking CTLA4/ligand interaction by Ipilimumab had therapeutic efficacy in a single case report [74]. CD28-ICOS fusion does not alter the CD28 molecule but may enhance its expression [73].

*CD28* was also affected by tandem duplications of 2q33.2 segments containing *CD28*, *CTLA4*, and *ICOS* and focal gains including high-level amplifications [23,80].

In the PI3K pathway, mutations occur in PIK3 elements encoding the regulatory subunits PIK3R1, PIK3R5, or the catalytic subunit PIK3CA in 7% of AITL and Tfh-like PTCL [2] and 1% of CTCL [28]. Following CD28 activation, PIK3R1(p85) binds the YXXM motif of CD28, and then activates the PIK3CA (p110a) catalytic subunit [10]. PIK3R1 bears the most mutations with K141R missense mutation affecting the r-GAP domain, Q475P/T576A and G680S/V704M affecting iSH2 and the second SH2 domains respectively [2]. A259V mutation maps to the linker region of PIK3R5, while the L1001P point mutation affects the PIK3CA kinase domain [2]. This mutation may enhance the catalytic subunit activity or increase PIK3R1 binding to CD28 [81].

The major mediator of PI3K-AKT signaling, PDK1, is encoded by *PDPK1* [82,83] (16p13.3), which is mutated in 5.9% of AITL [2]. Three missense mutations, including R324Q, P340Q, and T402M, located in or near the kinase domain, may exhibit an activating effect [84].

PI3K inhibition could be active against multiple T-cell lymphoid malignancies. The clinical activity of the PI3Kδ/γ inhibitor duvelisib in patients with CTCL and PTCL was reported and the overall response rates were 50.0% and 31.6%, respectively [85]. Tenalisib (RP6530), a dual PI3Kδ/γ inhibitor was evaluated in a phase I/Ib study in patients with relapsed or refractory T-cell lymphoma with an ORR of 45.7% [86]. TQ-B3525, a new PI3Kα/γ inhibitor is being evaluated for efficacy and safety in subjects with relapse/refractory PTCL (NCT04615468).

## 5. Differences in the Mutational Landscape between the Two Subtypes of PTCL-NOS

Based on gene expression signatures, PTCL-NOS has been delineated into two subgroups designated as GATA3 and TBX21, with different biological and prognostic features [87]. The GATA3 subtype showed type 2 helper T cell (Th2) characteristics with high expression of GATA3 and its target genes (*CCR4*, *IL18RA*, *CXCR7*, *IK*), whereas the TBX21 subtype showed Th1 features with higher expression of TBX21, EOMES, and their target genes (*CXCR3*, *IL2RB*, *CCL3*, *IFNG*). A subset of the TBX21 cases appears to have a cytotoxic phenotype with a poorer outcome. Significant enrichment of IFN α/β/γ-regulated gene signatures and NF-κB pathway signatures were observed in the TBX21 subtype compared with the GATA3 subtype that showed marginal enrichment for mTOR- and MYC-related gene signatures and significant enrichment of PI3K-induced gene signatures [87].

The genomic copy number analysis and targeted sequencing revealed distinct CN abnormalities and oncogenic pathways in PTCL-GATA3 and PTCL-TBX21 [80]. PTCL-GATA3 exhibited a high frequency (23%) of loss or mutation of tumor suppressor genes (*TP53*, *PTEN*, *FAS*, *CDKN2A/B*, and *PRDM1*) targeting the CDKN2A/B-TP53 axis and PTEN-PI3K pathways. Co-occurring gains/amplification of STAT3 and MYC are also characteristic of PTCL-GATA3 [80]. The PTCL-TBX21 subgroup had fewer CNAs (8%), primarily targeting cytotoxic effector genes (*CD244*, *CD247*, and *FASLG*), cell cycle regulator genes (*TP63*, *TPRG1*), and was enriched in mutations of genes regulating DNA methylation (*TET1*, *TET3*, and *DNMT3A*) [80]. CNAs affecting JAK-STAT (*SOCS1* and *JAK3*), PI3K-AKT (*ITPR3* and *ITPKB*), and T-cell signaling pathway (*PLCG1*, *PTPRC*, *FYN*, and *VAV1*) were common in both subgroups with no significant difference [80].

Watatani et al. [45] reported a series of PTCL-NOS cases with or without the Tfh phenotype. They identified a number of novel recurrently altered genes, including *KMT2C*, *SETD1B*, *YTHDF2*, and *PDCD1*. Using hierarchical clustering, they identified a cluster (cluster 1) that is enriched in cases with both *TET2* and *RHOA* mutations, with a subset harboring *IDH2* mutations as well, indicative of AITL and other Tfh-like lymphomas. [45] The PTCL-NOS cases in cluster 2 exhibited high genetic complexity and a high frequency of abnormalities in *TP53* and/or *CDKN2A*. This cluster also showed many alterations that may affect immunoregulatory functions such as *MHCI* or *MHCII*, *CD58*, *PDCD1*, and *FAS*, and the patients have the worst prognoses. Many of the molecular and clinical characteristics of cluster 2 are similar to those described for the PTCL-GATA3 subtype and they may contain largely overlapping patient populations [45,88]. It was reported that TP53 regulates CD4+ T-cell proliferation. Downmodulation of p53 by the induction of MDM2 upon TCR stimulation is critical for T-cell proliferation. It is possible that loss of *TP53* unleashes the cells from this control, making them more likely to respond to proliferation signals without proper TCR/CD28 activation. Loss of *TP53* may also have other oncogenic functions that need to be elucidated.

## 6. Future Perspective

Genetic alterations in genes related to TCR co-stimulation and signaling are detected in nearly all subtypes of PTCL entities. Mutations in specific genes occur at variable frequencies in different entities. Currently, there is no general recommendation to perform NGS on every case of PTCL. This may change in the future with the development of more targeted therapies directed against specific molecular abnormalities. There are, however, situations where genetic findings may be helpful in the diagnosis and classification of T-cell lymphoma when routine approaches are unable to make a definitive diagnosis. For example, the IDH R172 mutation is quite specific for AITL and the RHOA G17V mutation is very supportive of the diagnosis of Tfh-associated lymphoma if present. ITK-SYK fusion may be a relatively specific marker for follicular T-cell lymphoma [89] if further validated. PTCLs are often challenging to diagnose and classify, even for hematopathologists. A recent approach using gene expression signatures could help to provide a more robust and uniform diagnosis (Amador et al., JCO in press) and could be very useful in the stratification of patients in clinical trials. However, it is unclear what determines the preferred usage of a particular mutation in certain entities, such as the almost exclusive presence of *RHOA^G17V^* mutations in Tfh-related PTCL. Most of the altered genes reported harbor activating mutations promoting TCR/CD28 signaling and thus induce one or more downstream pathways such as MAPK, NFAT, and NF-κB and PI3K/AKT/mTOR pathways. However, TCR signaling needs to be within a certain range and excessive signaling could induce apoptosis. The mechanisms of action of these mutations and how they would interact with CD3/CD28 signaling and tumor microenvironment signals in the pathogenesis of PTCL need further investigation. The generation of appropriate animal models may help in these studies. There are few authentic cell lines derived from Tfh like PTCL and the PTCL-GATA3/TBX21 tumors, hindering in vitro investigation. It is now possible to modify normal T-cells in vitro by knockin of mutated variants and/or transduction of mutant genes. These modified cells could be very helpful in understanding how these mutants affect TCR signaling without the noise due to the presence of many other abnormalities in a tumor. Knockin of a mutant gene is preferable to viral transduction as the expression of the gene is under normal control, avoiding overexpression associated with viral vectors. The invention of CRISPR screening technologies has made it possible to identify complex interactions between cooperative mutations and reveal the driver gene in oncogenesis. Based on a better understanding of the molecular mechanism, potential therapeutic targets may be more properly predicted.

Given the prominent role of abnormal TCR signaling in tumor pathogenesis, several kinase inhibitors have been tested in preclinical experiments and clinical trials (listed in Table 2), including the SRC/ABL kinase inhibitor dasatinib [2,90], anti-CCR4 antibody, Mogamulizumab [48], bromodomain inhibitor JQ1 [43], ITK inhibitor CPI-818, dual SYK/JAK inhibitor Cerdulatinib [59], and PI3K inhibitors such as duvelisib [85], tenalisib [86], and TQ-B3525. The CTLA4 antibody [74] could potentially be active against tumors with CTLA-CD28 fusion. PKC inhibitor MS-553 and MALT1 inhibitor JNJ-67856633 which are now evaluated for B-cell lymphoma in clinical trials, as well as IKK inhibitors BMS-345541 and IKK-16 [51], may also be effective for T-cell lymphoma with *PRKCB*, *CARD11* alterations, or FYN-TRAF3IP2 fusion. Some of the clinical trials have reported promising results, but often with different efficacy among different PTCLs. It is, therefore, important that patients are well stratified and characterized to properly interpret the results of the trials. Epigenetic alterations are prominent in PTCL, particularly in Tfh-related tumors. It is important to determine how these epigenetic changes interact with TCR signaling alterations and what the role of combining epigenome modifying drugs (HDACi, DNMTi, EZH1,2i) with the different kinase inhibitors is. With better mechanistic understanding of PTCL, novel drugs, drug combinations, and new strategies may be identified to target abnormal TCR signaling and epigenetic changes more precisely and synergistically to achieve better results for this group of patients with poor outcomes.

## Figures and Tables

**Figure 1 cancers-14-03716-f001:**
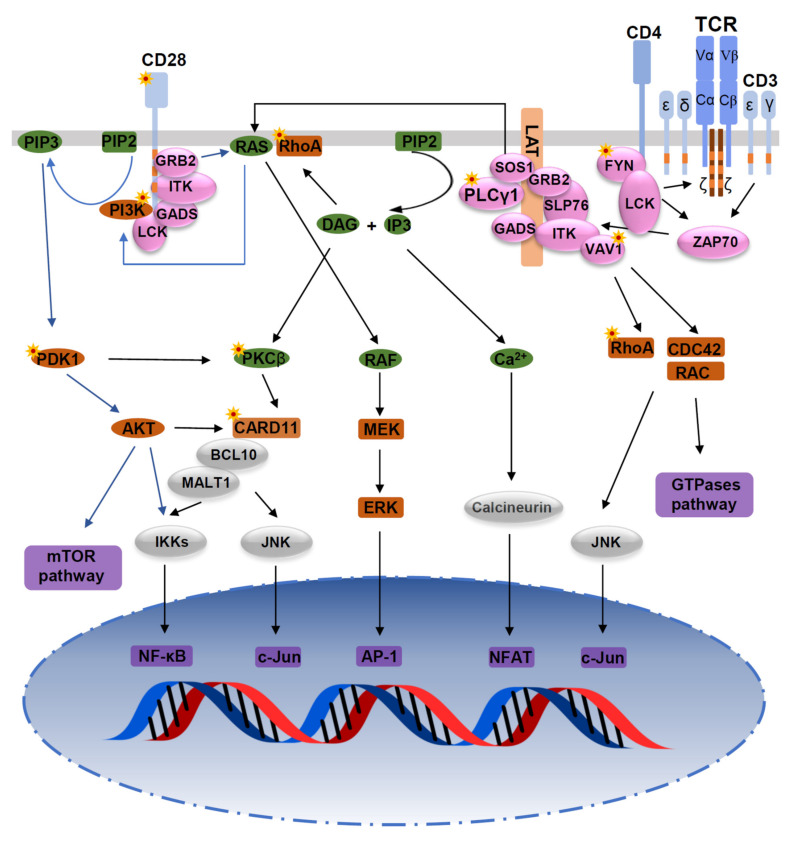
Mutations in the TCR signaling pathway. Mutations of TCR signaling-related genes in PTCL. The intracellular pathways after TCR ligation and costimulatory activation were reconstructed from published studies. From left to right: (1) PI3K pathway after CD28/TCR-dependent FYN phosphorylation and ultimately resulting in activation of mTOR and NF-κB pathways; (2) AP-1/MAPK pathway that comprises MALT1-induced JNK activation, and PLCγ1-GRB2/SOS–induced MAPK activation; (3) NF-κB/NFAT pathway proximally initiated by ITK-dependent PLCγ1 activation; and (4) GTPase-dependent pathway, including RHOA, responsible for cytoskeleton remodeling upon costimulatory/TCR activation. Asterisks indicate mutations affecting the respective molecules.

**Figure 2 cancers-14-03716-f002:**
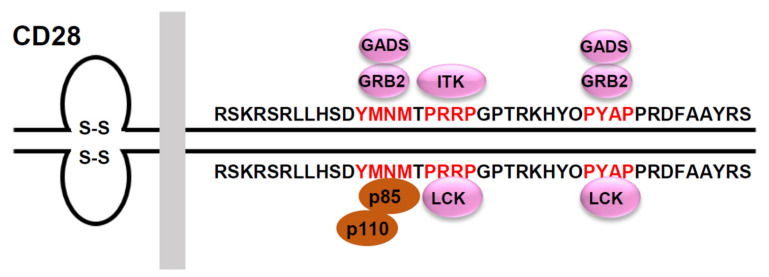
Signaling motifs in the cytoplasmic tail of the human CD28 and its binding partners. The human CD28 possesses a 41 amino acid-long cytoplasmic tail that includes three potential protein-protein interaction motifs (highlighted in red). The phospho-Tyr173 within the YMNM motif serves as a docking site for the SH2-containing proteins, p85, GRB2 and GADS. The PRRP motif can interact with the SH3 domain of ITK and LCK. The PYAP motif can interact with the SH3 domain of GRB2, GADS, and LCK.

**Figure 3 cancers-14-03716-f003:**
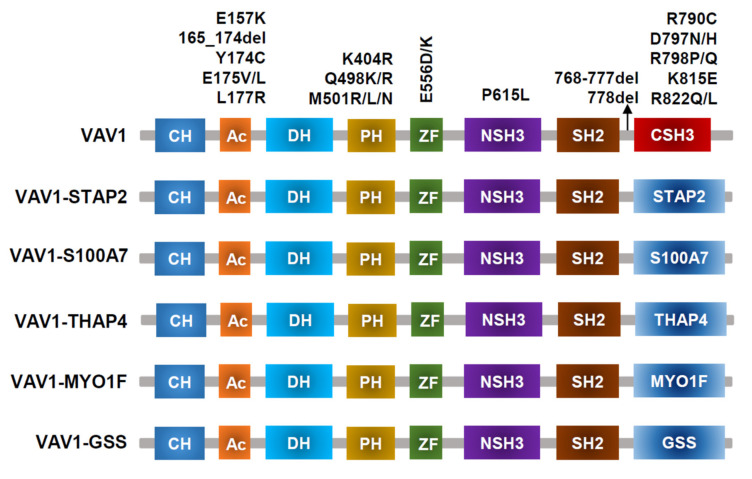
VAV1-mutant proteins resulting from nonsynonymous mutations, in-frame deletions, and fusion with various partners identified in PTCL-NOS, AITL, ALCL, and ATLL.

**Figure 4 cancers-14-03716-f004:**
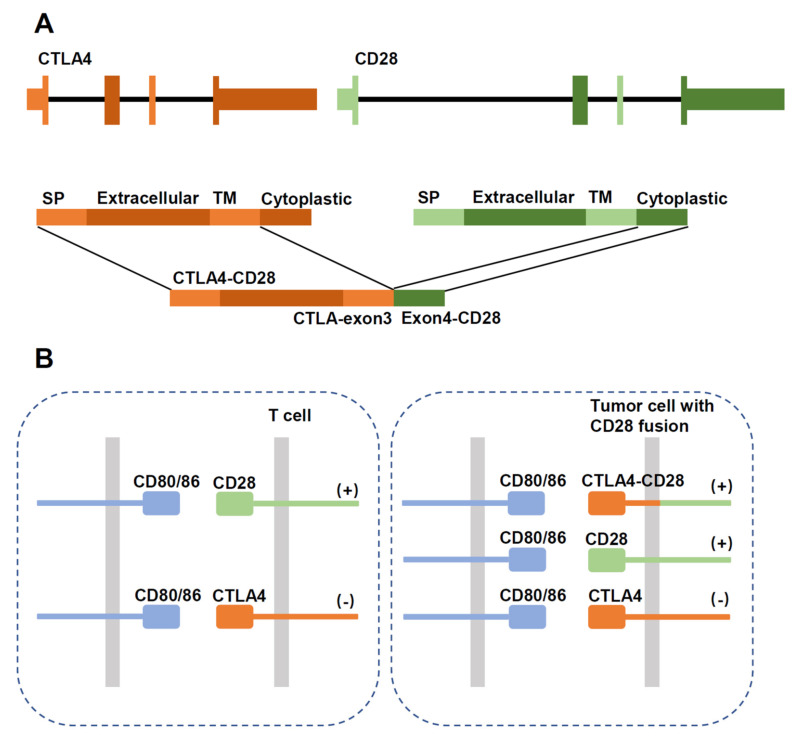
Schematic diagram showing structure of the CTLA4-CD28 gene fusion. (**A**) Schematic diagram of the gene fusion. SP: signal peptide; TM: transmembrane region. (**B**) In normal T cells, activation of CD28 stimulates proliferation, which is inhibited by CTLA4 signaling. In tumor cells expressing the fusion protein, CTLA4 activation would aberrantly stimulate proliferation through the intracellular CD28 domain.

**Table 1 cancers-14-03716-t001:** Characteristics of *RHOA* mutations in PTCLs.

Amino Acid Change	Mutation Type	Domain	Function	Affected Subtypes	Ref.
G14V	missense	GTP binding	Gain of function	ATLL	[23]
C16R	missense	GTP binding	Gain of function	ATLL, CTCL	[25,28]
C16F/G	missense	GTP binding	Gain of function	ATLL	[23]
C16L	missense	GTP binding	Not validated	ATLL	[23]
G17E	missense	GTP binding	Gain of function	ATLL	[25]
K18N	missense	GTP binding	Gain of function	AITL	[2]
K118E	missense	GTP binding	Gain of function	ATLL, CTCL	[23,28]
K118T	missense	GTP binding	Loss of function	ATLL, CTCL	[23,28]
A161P	missense		Gain of function	ATLL, CTCL	[23,28]
A161V	missense		Gain of function	ATLL	[23,28]
G17V	missense	GTP binding	Loss of function	ATLL, AITL, PTCL-NOS	[23,25]
N117I	missense	GTP binding	Loss of function	ATLL, CTCL	[25,28]
N117K	missense	GTP binding	Not validated	CTCL	[28]
T19I	missense	GTP binding	Not validated	ATLL	[25]
A56V	missense		Not validated	ATLL	[25]
R68L	missense		Not validated	ATLL	[25]
R68C	missense		Not validated	CTCL	[28]
R70K	missense		Not validated	CTCL	[28]
C83Y	missense		Not validated	ATLL	[25]
D120Y	missense		Not validated	PTCL	[25]
K162E	missense		Not validated	ATLL	[25]
K188Q	missense	Hypervariable region	Not validated	ATLL	[25]

**Table 2 cancers-14-03716-t002:** Genetic alterations related to TCR signaling pathway in PTCLs.

Genes	Affected Subtypes	Coexistent Mutations	Presumed Mechanisms	Prognostic Value	Potential Drugs
*RHOA*	50–71% in AITL [2,25,26] 8–18% of PTCL-NOS [25] 5% of CTCL [28] 8% of ATLL [23]	*TET2* [2]	RHOA^G17V^ may bind to VAV1 and augment its adaptor function, resulting in acceleration of TCR signaling, enhancing RAC1 and PI3K signaling, and enrichment of cytokine and chemokine-related pathways, and Tfh polarization [19,31]	Poor OS and PFS in RHOA^G17V^ -mutated AITL [38]	A pan-Src inhibitor PP2 and Src inhibitor Dasatinib [31]
*VAV1*	18% in ATLL [23] 11% in ALCL, 7% to 11% in PTCL-NOS, 5% in AITL	May be negatively correlated with *RHOA mutation* [2]	Alterations disrupt VAV1 autoinhibition and lead to spontaneous activation of TCR signaling and increased NFAT and NF-κB activity [46,47]	Not reported	Anti-CCR4 antibody Mogamulizumab [48], bromodomain inhibitor JQ1 [43]
*FYN*	4% in PTCL [25]	FYN-TRAF3IP2 is devoid of RHOA ^G17V^ [51]	FYN^L174R/R176C/Y531H^ as activating mutations with abrogation of the auto-inhibitory loop and increased levels of FYN phosphorylation. FYN-TRAF3IP2 leads to aberrant NF-κB signaling downstream of TCR activation [51]	Not reported	IKK inhibitors BMS-345541 and IKK-16 [51]
*LCK*	reported in a single AITL patient [2]	Not reported	Not reported	Not reported	Not reported
*ITK-SYK*	17% in PTCL-NOS [54]	Not reported	ITK-SYK activates IL2RG/JAK3/STAT5 signaling pathway [56]	Not reported	ITK inhibitor CPI-818(NCT03952078); JAK-SYK inhibitor Cerdulatinib [59]
*PLCG1*	36% in ATLL [23] 14.1% in AITL and other Tfh-cell-like lymphomas [2] 1.3–21% in CTCL [28,60,62]	no specific association	promoting MALT1 cleavage and NFAT activity [2]	Not reproted	Not reported
*PRKCB*	33% in ATLL [23] 0.4% in CTCL [28]	a significant positive correlation with *CARD11* mutations in ATLL [23]	resulting in PKCβ activation and the enhancement of the NF-κB pathway [23]	A poor prognostic factor in aggressive ATLL [91]	PRKCB inhibitor MS-553 [72]
*CARD11*	24% of ATLL [23] 3.5% of AITL and Tfh like PTCL [2] 5.4% of CTCL [28]	a significant positive correlation with *PRKCB* mutations in ATLL [23]	*CARD*^E626K^ affects the auto-imhibition of PKC and leads to the activation of CARD11 and increase of NF-κB activity [23,71].	Not reported	MALT1 inhibitor JNJ-67856633 (NCT03900598)
*CD28*	16.4% in AITL [2,73] 9–10.8% in ATLL [23] 3.6% in CTCL [28]	no specific association,	CD28 ^D124V^ has a higher affinity for its ligand CD86 than CD28 WT [73]. CD28 ^T195P^ has a higher affinity for GRB2 and GADS/GRAP2 than CD28 ^WT^ [76]. The CTLA4-CD28 fusion protein has enhanced ability to promote T-cell activation.	CD28 overexpression and mutation as a poor prognostic factor [73,92]	Ipilimumab [74] for CTLA4-CD28 fusion
PI3K elements	7% of AITL and Tfh-like PTCL [2] 1% of CTCL [28]	mutations common in AITL	These mutants may enhance the catalytic subunit activity or increase PIK3R1 binding to CD28 [81].	Not reported	Duvelisib [85], Tenalisib [86], Tenalisib with romidepsin(NCT02017613) TQ-B3525 (NCT04615468)

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
