# Peer review of "Mutations Affecting Genes in the Proximal T-Cell Receptor Signaling Pathway in Peripheral T-Cell Lymphoma"

_cancers, 2022, doi:10.3390/cancers14153716_

Round 1

Reviewer 1 Report

This is a well written and interesting manuscript for researcher in the field of  PTCL molecular pathology .  However, it should outline a more comprehensive vision, appealing also for less specialised readers.

Therefore, I suggest to carry out tables and graphs to deliver an overall view of the topic and to connect these mutations  to histologic subtypes, therapies, coexistence of other mutations, therapeutic targets, etc . In addition,  the future perspective should be further developed. I also  suggest to outline which should be an updated  molecular workup at diagnosis and how molecular diagnosis could change future, treatment approaches, which trials are ongoing?

Tables and graphs should summarize the following

1)for each mutation which are the affected  histologic subtypes and their relative frequency

2) if  it is a presumed  driver mutation and which is the supposed pathogenetic mechanism

3) the coexistence  and the  possible interactions with other known  mutations

4) which  are  biological/targeted drugs which showed activity in animal models/patients harbouring the mutation

5) if the mutation is prognostically relevant

Author Response

Response to Reviewer 1 Comments

Reviewer1:

This is a well written and interesting manuscript for researcher in the field of  PTCL molecular pathology .  However, it should outline a more comprehensive vision, appealing also for less specialised readers.

Therefore, I suggest to carry out tables and graphs to deliver an overall view of the topic and to connect these mutations  to histologic subtypes, therapies, coexistence of other mutations, therapeutic targets, etc . In addition,  the future perspective should be further developed. I also  suggest to outline which should be an updated  molecular workup at diagnosis and how molecular diagnosis could change future, treatment approaches, which trials are ongoing?

Reply: Thanks for the insightful comments. We discuss further the molecular workup at diagnosis in the revised paper (lines 387-395). The ongoing clinical trials related to the TCR signaling pathway are limited. We listed potential targeted therapies in lines 415-423. Hopefully, with a better understanding, more clinical trials may be carried on in PTCL.

Tables and graphs should summarize the following

1)for each mutation which are the affected  histologic subtypes and their relative frequency

2) if  it is a presumed  driver mutation and which is the supposed pathogenetic mechanism

3) the coexistence  and the  possible interactions with other known  mutations

4) which  are  biological/targeted drugs which showed activity in animal models/patients harbouring the mutation

5) if the mutation is prognostically relevant

Reply: Thanks very much for the positive comments.

We add a table (Table2. Line 344) to summarize the affected subtypes, frequency, coexistent genetic alterations, and prognostic value of the mutations. As to the role and pathogenetic mechanism of these mutants, they all affect the proximal TCR signaling pathway in different ways as discussed in the text, but due to the limited mechanistic understanding, how precise does each one of them mediate transformation and how this is influenced by CD3/CD28 signaling, tumor microenvironmental signals and epigenomic alterations still needs further investigation in the future.

Reviewer 2 Report

The authors first take an overview of TCR signaling pathways and propose the association of chronic TCR signaling and PTCLs. Then, the authors summarize frequent mutations found in genes involved in T cell proximal signaling pathways in the context of peripheral T-cell lymphomas. Overall, this is a very thorough and well-written review.

What is the main question addressed by the research? The aim of this review is to summarize and to discuss the functional impact of the genomic alterations of genes involved in proximal TCR signaling pathway identified in different subgroups of PTCL patients.    Is it relevant and interesting? What does it add to the subject area compared with other published material? How original is the topic?   Because molecular abnormalities in PTCLs frequently target genes encoding proteins with important roles in normal T-cell activation and also as this paper pointed out chronic antigenic stimulation and persistent TCR signaling is often observed in PTCLs, articles specifically focusing on summarizing genetic alterations of proximal T cell signaling would help better understand the patheogeneis and biology of the disease across different subtypes and may provide a new perspective to identify potential targets for the treatment.    To my knowledge, most review papers tend to review frequent genetic alterations in the context of particular T-cell lymphoma.  In this paper, they try to understand this disease from the perspective of persistant TCR signaling, mechanisms by which essentially all types of T cell lymphoma requires.  Chronic T cell signal 1 activation in the absence of signal 2 would make T cells anergy or death, but mechanisms of how T cells overcome this through modifying proximal TCR signaling and allow normal T cells become malignancy may be shared across different T cell lymphoma subtypes. This review provided a nice summary of those alteration and thus is of interest to the academic as a whole.      Is the paper well written? Is the text clear and easy to read? To me, this paper is well written and organized. In section 2, the authors reviewed TCR signaling pathways and in the latter sections they discuss how those signaling went awry as a result of genetic alterations in the context of T cell lymphomas. 

Author Response

Thank you very much for the positive comments!

Round 2

Reviewer 1 Report

The Authors did not properly addressed my previous comments. In addition the new table  and the few lines added in the "future perspective " section have several grammar mistakes 

There is still  a  lack/misleading  informations about potentially active drugs in the setting of specific mutations ( i.e. no mention about such as dasatinib in PTCL harbouring RHOA, FYN  mutations, Duvelisib and TQ-B3525 in PIK3 mutations or Ipilimumab in ICOS-CD28 fusion, etc)

Also the few lines about molecular diagnosis are lousy

In my opinion , the revised parts are overall inaccurate and poor

Author Response

Dear Dr. Ugrinović and reviewer,

Re to “Mutations Affecting Genes in the Proximal T-cell Receptor Signaling Pathway in Peripheral T-cell Lymphoma” (Manuscript ID: cancers-1746214)

We are sorry that the first revision did not address the comments of reviewer 1 clearly and satisfactorily. We have revised the paper including more literature review as well as more discussions on potential drugs and ongoing clinical trials. We hope this version has addressed all the concerns of the reviewer.

Our response to the reviewer’s comments is attached.

Best regards,

Xiaoqian Liu and John Chan

Point-to-point response

Reviewer1:

“The Authors did not properly addressed my previous comments. In addition the new table  and the few lines added in the "future perspective " section have several grammar mistakes”

Reply:

We listed the comments in round 1 below and provided revised responses to them as detailed below.

“This is a well written and interesting manuscript for researcher in the field of  PTCL molecular pathology .  However, it should outline a more comprehensive vision, appealing also for less specialised readers.

Therefore, I suggest to carry out tables and graphs to deliver an overall view of the topic and to connect these mutations  to histologic subtypes, therapies, coexistence of other mutations, therapeutic targets, etc . In addition,  the future perspective should be further developed. I also  suggest to outline which should be an updated  molecular workup at diagnosis and how molecular diagnosis could change future, treatment approaches, which trials are ongoing?”

Reply: 

First, as suggested by the reviewer, for a better and more comprehensive revision, we have added a comprehensive table (Table 2 in line 380) that summarizes the incidences of these mutations in different histologic subtypes, common coexistent mutations, presumed mechanisms, prognostic values, and potential drugs if reported. We have mentioned in the text, drugs based on the presumed mechanisms of the mutant genes, that have been evaluated and published such as dasatinib and duvelisib, as well as drugs that are being evaluated such as CPI-818 and TQ-B3525 obtained from the clinical trials website. This information has now been added to Table 2. We have also discussed the treatment strategies for PTCL with certain mutations in detail in the “4. Activating mutations in genes related to TCR signaling in PTCLs”(lines 177-184 for RHOA, lines 209-215 for VAV1, lines 238-240 for FYN, lines 259-265 for ITK, lines 330-334 for PRKCB and CARD11, lines 372-378 for PI3K and Ipilimumab for ICOS-CD28 fusion lines 354-355) and at the end of this paper (lines 455-467).

We discussed further the molecular workup at diagnosis in the revised paper (lines 424-435). Currently, there is actually no general recommendation to perform NGS on patients with PTCL. This may change in the future with the development of more targeted therapy directed against specific molecular abnormalities. However, in certain situations, genetic findings may be helpful in the diagnosis and classification of T-cell lymphoma when routine approaches are unable to make a definitive diagnosis. We have added this information to the manuscript. A recent approach using gene expression signatures could help to provide a more robust and uniform diagnosis (Amador et al JCO, in press) that may be useful in the stratification of patients in clinical trials and this is now added to the manuscript.

“There is still  a  lack/misleading  informations about potentially active drugs in the setting of specific mutations ( i.e. no mention about such as dasatinib in PTCL harbouring RHOA, FYN  mutations, Duvelisib and TQ-B3525 in PIK3 mutations or Ipilimumab in ICOS-CD28 fusion, etc)”

Reply :

We have discussed more extensively on treatment strategies in the section “3. Chronic antigenic stimulation and persistent TCR signaling in PTCLs” as detailed above.

“Also the few lines about molecular diagnosis are lousy”

We discussed further the molecular workup at diagnosis in the revised paper (lines 424-435) as detailed above. That should contain the current information on diagnosis that is relevant to the subject matter of this review. A general discussion on the diagnosis/molecular diagnosis of PTCL, while interesting and important, is beyond the scope of this review.

Round 3

Reviewer 1 Report

the paper was well revised and I reccomend its pubblication in Cancers.

I suggest, that the implementations are also summarized in the abstract section ( there is no mention of targeted therapies in it)

Author Response

Dear reviewer,

Thank you for your comments. We summarized the implementations in the parts of "simple summary" and "abstract" in the latest version attached.